# Effect of Exercise Training on Body Temperature in the Elderly: A Retrospective Cohort Study

**DOI:** 10.3390/geriatrics6010003

**Published:** 2021-01-01

**Authors:** Koichiro Matsumura, Toshiji Iwasaka, Satoshi Mizuno, Ikuko Mizuno, Hikaru Hayanami, Kiyoshi Sawada, Junji Iwasaka, Kotaro Takeuchi, Toshimitsu Suga, Tetsuro Sugiura, Ichiro Shiojima

**Affiliations:** 1Department of Medicine II, Kansai Medical University, Osaka 5708507, Japan; iwasakaj@hirakata.kmu.ac.jp (J.I.); sugiurat@mc.kmu.ac.jp (T.S.); shiojima@hirakata.kmu.ac.jp (I.S.); 2Department of Internal Medicine, Tsurumi Ryokuchi Hospital, Osaka 5700044, Japan; iwasaka@seisuikai.net (T.I.); hamadas@seisuikai.net (S.M.); ikuko3896@seisuikai.net (I.M.); hayanami@seisuikai.net (H.H.); sawada@seisuikai.net (K.S.); 3Department of Physical Medicine and Rehabilitation, Kansai Medical University, Osaka 5708507, Japan; kotaro0307@gmail.com (K.T.); sugat@hotmail.co.jp (T.S.)

**Keywords:** body temperature, elderly, exercise, skeletal muscle mass

## Abstract

Background: This study evaluated the effect of exercise training on body temperature and clarified the relationship between body temperature and body composition in the elderly. Methods: In this retrospective cohort study, a total of 91 elderly participants performed aerobic and anaerobic exercise training twice a week for 2 years. Non-contact infrared thermometer and bioelectrical impedance analysis were performed at baseline and at 2 years. Results: Mean age of study participants was 81.0 years. The participants were divided into two groups by baseline body temperature of 36.3 °C; lower body temperature group (n = 67) and normal body temperature group (n = 24). Body temperature rose significantly after exercise training in the lower body temperature group (36.04 ± 0.11 °C to 36.30 ± 0.13 °C, *p* < 0.0001), whereas there was no significant difference in the normal body temperature group (36.35 ± 0.07 °C to 36.36 ± 0.13 °C, *p* = 0.39). A positive correlation was observed between the amount of change in body temperature and baseline body temperature (r = −0.68, *p* < 0.0001). Increase in skeletal muscle mass was an independent variable related to the rise in body temperature by the multivariate logistic regression analysis (odds ratio: 4.77, 95% confidence interval: 1.29–17.70, *p* = 0.02). Conclusions: Exercise training raised body temperature in the elderly, especially those with lower baseline body temperature.

## 1. Introduction

Body temperature is one of the major physiological parameters related to health condition, which differs by age and elderly people have a lower body temperature compared to young adults [1]. The decrease in body energy expenditure, skeletal muscle mass and physical activity in the elderly attenuates to lower body temperature [2,3,4], which contributes to inactivate immune function against infection and cancer [5,6,7,8,9]. Exercise training, including aerobic and anaerobic, is effective to increase body energy expenditure and skeletal muscle mass. Although exercise training contributes to raising body temperature, little is known about the relationship between exercise training and body temperature. Accordingly, we investigated the effect of exercise training on body temperature and clarified the relationship between body temperature and body composition in the elderly.

## 2. Materials and Methods

### 2.1. Study Population

This single center retrospective cohort study included 91 consecutive elderly participants (≥70 years old) between December 2015 and December 2016. All participants who had exercise training twice a week for 2 years. Participants with interruption of exercise training over 2 years were excluded in this analysis. We obtained written informed consent from all participants. The study protocol was approved by the ethics committee of Tsurumi Ryokuchi Hospital (No. 2018R1). The investigation conformed with the principles outlined in the Declaration of Helsinki of 1975. Baseline clinical characteristics including age, sex, hypertension, diabetes mellitus, orthopedic disease and previous cerebral infarction were extracted from the medical record. Participants were categorized into 3 types of activity of daily living: (1) independent outdoor walking; (2) independent indoor walking, or (3) indoor walking with assistance.

### 2.2. Evaluation of Body Temperature and Body Composition

Body temperature was measured before each exercise from the participant’s forehead using a non-contact infrared thermometer (Thermo Phrase MT-500, NISSEI, Gunma, Japan) after a 15 min rest in a room maintained at 20 °C environmental temperature. Average body temperature for one month was evaluated to minimize individual body temperature variation at baseline and at 2 years. Participants were divided into 2 groups by baseline body temperature of 36.3 °C which is a standard body temperature in the elderly by previous systematic review: lower body temperature group (<36.3 °C) and normal body temperature group (≥36.3 °C) [1]. Body composition including body weight, metabolic rate, skeletal muscle mass and body fat mass were measured at baseline and at 2 years using bioelectrical impedance analysis (InBody 430, InBody Japan, Tokyo, Japan) [10].

### 2.3. Exercise Protocol

Exercise training consisted of aerobic and anaerobic exercise. After measuring body temperature, beginning with a warm-up stretch for 15 min, participants performed aerobic exercise; 10 min of cycling with an aero bike (AB158EXI, Konami Sports Life Co., Ltd., Kanagawa, Japan), and 10 min of stair ascent and descent training (SP-100, SAKAI Medical Co., Ltd., Tokyo, Japan). Aerobic exercise intensity was determined according to heart rate and Borg’s scale 11–13 during exercise as referenced by previous report [11]. Participants with indoor walking with assistance were excused from stair ascent and descent training. Anaerobic exercise consisted of leg press, leg extension, seated rowing, hip adduction, and torso flexion using exercise machine (COP 902F-906F, COP-STEP, SAKAI Medical Co., Ltd., Tokyo, Japan). Anaerobic exercise intensity was determined according to 40 to 60% of 1 repetition maximum as referenced by previous report [12]. All anaerobic exercise was attempted at 2 to 3 sets with 3 to 5 min interval among each set. Exercise intensity and volume were not changed throughout 2 years. All the exercise procedures were assisted by physical therapists.

### 2.4. Statistical Analysis

Continuous variables are presented as means with standard deviations and categorical variables as number of total (percentages). Differences between the 2 groups were analyzed using the Student’s *t*-test for continuous variables and the Chi-square test for categorical variables. To verify body temperature variation of a non-contact infrared thermometer, Bland-Altman plot was conducted. The upper and lower limits of agreement were set at 2 standard deviations from the mean. Differences between baseline and 2-year data were conducted using the paired *t*-test. Correlation between 2 variables was determined by the linear regression analysis. Multivariate logistic regression analysis using 6 variables was performed. Categorical variables were subdivided as skeletal muscle mass; increase of >0 kg or ≤0 kg, and metabolic rate; increase of >0 kcal or ≤0 kcal. JMP 14.2.0 software (SAS Institute Inc., Cary, NC, USA) was used for all statistical analyses. A *p*-value < 0.05 was considered significant.

## 3. Results

### 3.1. Patient Characteristics

Mean age of study participants was 81.0 years and mean body temperature was 36.1 ± 0.2 °C. Baseline clinical characteristics of the lower body temperature group (n = 67) and the normal body temperature group (n = 24) are shown in Table 1. Body mass index and body fat mass were significantly lower in lower body temperature group. The distribution of activity of daily living was comparable between the two groups.

### 3.2. Body Temperature Variation

To verify body temperature variation of a non-contact infrared thermometer, the difference between average body temperature and each body temperature in individual participant at baseline and at 2 years was evaluated (Figure 1). Bland-Altman plot revealed acceptable degree of agreement of body temperature variation using a non-contrast infrared thermometer was confirmed (bias 0.002, 95% confidence interval −0.006 to 0.010).

### 3.3. Change in Body Temperature

Figure 2 shows the change in body temperature from baseline to 2 years. Lower body temperature group showed a significant rise in body temperature at 2 years (36.04 ± 0.11 °C to 36.30 ± 0.13 °C, *p* < 0.0001), whereas there was no significant difference in the normal body temperature group (36.35 ± 0.07 °C to 36.36 ± 0.13 °C, *p* = 0.39).

Compared to normal body temperature group, lower body temperature group had a significantly larger change of body temperature from baseline to 2 years (lower body temperature: 0.26 ± 0.12 °C vs. upper body temperature: 0.01 ± 0.12 °C, *p* < 0.0001, Figure 3).

The number of changes in body temperature from baseline to 2 years had a strong correlation with baseline body temperature (Figure 4).

### 3.4. Relationship between Body Temperature and Change in Body Composition

Lower body temperature group had significantly higher incidence of participants with an increase in skeletal muscle mass after exercise compared to normal body temperature group, whereas there was no significant difference in metabolic rate and body fat mass between the two groups (Figure 5).

### 3.5. Relationship between Status of Activity of Daily Living and Body Composition and Body Temperature

When participants were classified according to status of activity of daily living, incidence of increase in metabolic rate (56% vs. 44% vs. 39%, *p* = 0.36), skeletal muscle mass (40% vs. 44% vs. 35%, *p* = 0.82), body fat mass (66% vs. 50% vs. 48%, *p* = 0.25), and body temperature (80% vs. 78% vs. 96%, *p* = 0.19) were no significant differences among participants with independent outdoor walking, independent indoor walking, and indoor walking with assistance, respectively.

### 3.6. Variable Related to the Rise in Body Temperature

When multivariate logistic regression analysis was performed, increase in skeletal muscle mass after exercise emerged as an independent variable related to the rise in body temperature (odds ratio: 4.73, 95% confidence interval: 1.29–17.26, *p* = 0.02, Table 2).

## 4. Discussion

Body temperature is strictly regulated at the hypothalamus region because thermoregulation is crucial to human life [13]. Heat generation is consisted of metabolic heat production, skeletal muscle thermogenesis and physical activity. Elderly people have relatively low heat generation, impaired thermal perception and thermal regulatory response compared to young adults [4]. In addition, thermogenesis by meal intake is negatively affected by age because energy intake in the elderly tends to be lower than their energy expenditure, which results in a gradual decrease in metabolic rate [14]. However, variation of metabolic rate exists independent of age and gender. A small number study showed a significant correlation between oral body temperature and metabolic rate in healthy participants [2]. A cohort study of 18,630 participants aged 20–98 years showed a positive correlation between body mass index and body temperature both men and women in all generations [1]. These data indicate that diminished thermogenesis associated with lower metabolic rate and skeletal muscle mass cause decrease in body temperature in the elderly [15]. Moreover, reduction in meal intake accelerates weight loss and decreases skeletal muscle mass, resulting in further reduction in heat generation. In this study, incidence of participants with increase in metabolic rate was comparable between participants with lower and normal body temperature group. In contrast, increase in skeletal muscle mass was significantly higher in participants with lower body temperature compared to those with normal body temperature. Moreover, multivariate analysis indicated that the increase in skeletal muscle mass was an independent variable related to the rise in body temperature, whereas there was no significant association between increase in metabolic rate and rise in body temperature. This implies that skeletal muscle mass may be strongly attributable to a rise in body temperature rather than metabolic rate. Regarding this point, further investigation is needed by clinical and experimental studies.

In this study, lower body mass index and body fat mass were observed in lower body temperature group compared to normal body temperature. Brown adipose tissue has a thermogenic capacity and its main role is to preserve human body temperature [4]. Although amount of brown adipose tissue was not assessed in our study, lower body mass index and body fat mass may have contributed to lower the temperature in the lower body temperature group. However, body temperature rose significantly after 2 years of exercise training in the elderly with baseline body temperature <36.3 °C. There was a strong correlation between baseline body temperature and the amount of rise in body temperature, indicating that 2 years of exercise training resulted in a higher degree of body temperature elevation in lower body temperature group. Participants with an increase in skeletal muscle mass was significantly higher in the lower body temperature group than that in normal body temperature group. Moreover, increase in skeletal muscle mass was an independent variable related to the rise in body temperature. Therefore, we consider that skeletal muscle mass rather than metabolic rate was strongly associated with the rise in body temperature after 2 years of exercise training.

Several experimental studies have shown that rise in body temperature activates the immune system which contributes to protecting against infection and cancer [5,6,7,8]. CD8+ T cell activated by hyperthermia is capable of destroying virus-infected cells and cancer cells. Mace et al. investigated the activation of CD8+ T cell using mice with injected antigen and found accelerated generation and differentiation of CD8+ T cell in warmed mice with a 2 °C rise of body temperature [5]. Foxman et al. demonstrated a close relationship between function of interferon and body temperature [6,7]. More quick replicates and spread of common cold virus were observed in mouse airway cells in mice with lower body temperature due to impairment of interferon function. Although clinical significance of body temperature elevation was not evaluated in our study, rise in body temperature associated with increase in skeletal muscle mass after exercise training may have clinical benefits in the elderly with lower body temperature.

Three limitations of our study should be addressed. First, the study population was limited in number and retrospectively investigated. The findings of this study need to be confirmed prospectively in a larger population to determine the clinical significance of rise in body temperature in the elderly. Nevertheless, this is the first study to evaluate the changes in body temperature after exercise training in the elderly. Second, there is no data concerning muscle strength and functional testing such as grip strength and gait speed which is often used for sarcopenia diagnosis [16]. Further evaluation of the relationship between body temperature and muscle strength and function is needed. Third, this study did not include the control group without undergoing exercise training. Comparing change in body temperature and body composition between elderly participants with and without exercise training would be attributable to better understanding of the association between exercise training and body temperature.

## 5. Conclusions

Body temperature rose after 2 years of exercise training in the elderly, especially those with a lower baseline body temperature.

## Figures and Tables

**Figure 1 geriatrics-06-00003-f001:**
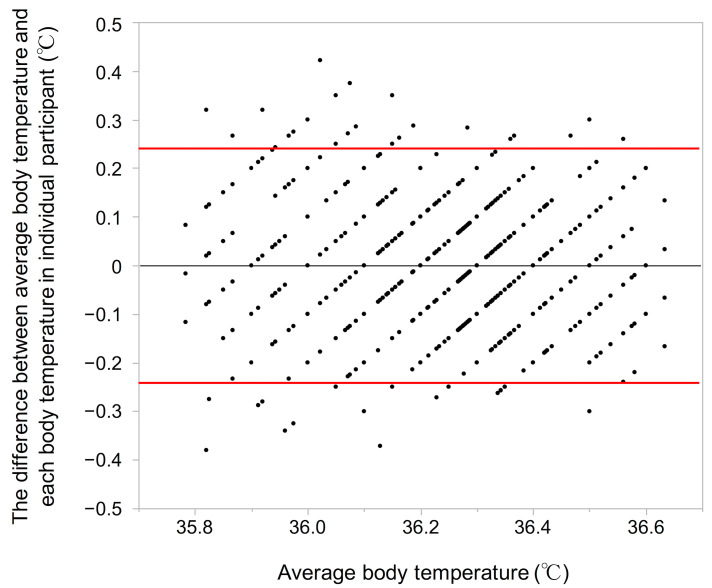
Bland-Altman plot assessing body temperature variation of a non-contact infrared thermometer between average body temperature and each body temperature in individual participant at baseline and at 2 years. The upper and lower limits of agreement were set at two standard deviations from the mean (red lines). Black line indicates mean difference.

**Figure 2 geriatrics-06-00003-f002:**
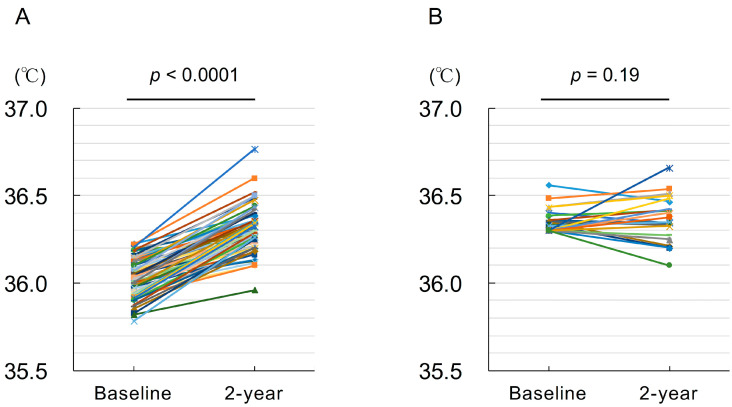
Change in body temperature after exercise training. Lower body temperature group (**A**) and normal body temperature group (**B**).

**Figure 3 geriatrics-06-00003-f003:**
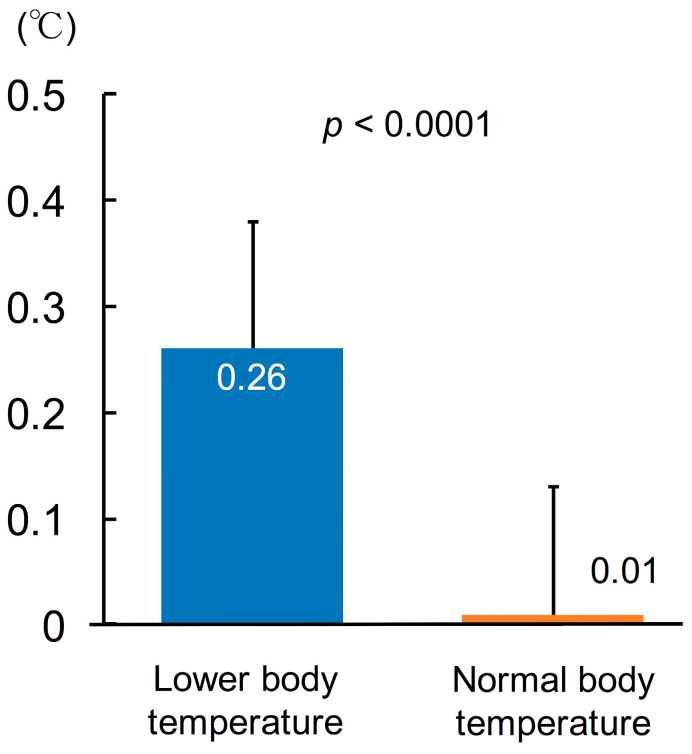
The amount of changes in body temperature after exercise training.

**Figure 4 geriatrics-06-00003-f004:**
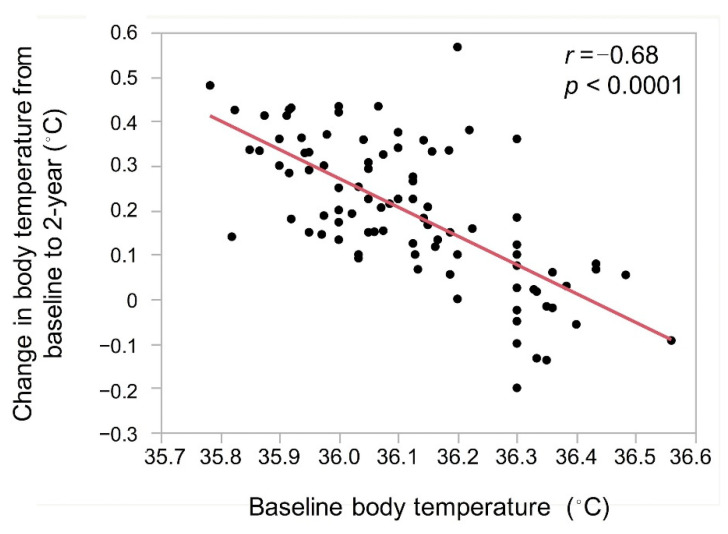
Correlation between change in body temperature after exercise training and baseline body temperature.

**Figure 5 geriatrics-06-00003-f005:**
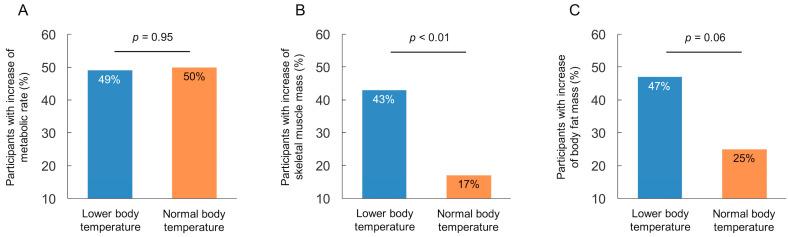
Comparison of participants with increase of metabolic rate (**A**), skeletal muscle mass (**B**), and body fat mass (**C**) after 2 years exercise training.

**Table 1 geriatrics-06-00003-t001:** Baseline clinical characteristics.

	Lower Body Temperature (n = 67)	Normal Body Temperature (n = 24)	*p* Value
Age (years)	81.3 ± 4.5	80.5 ± 3.6	0.49
Male	24 (36)	8 (33)	0.83
Body temperature (°C)	36.0 ± 0.1	36.4 ± 0.1	<0.0001
Comorbidity			
Hypertension	18 (27)	5 (21)	0.56
Diabetes mellitus	9 (13)	2 (8)	0.51
Orthopedic disease	35 (52)	14 (58)	0.61
Previous cerebral infarction	5 (7)	1 (4)	0.58
Activity of daily living			0.70
Independent outdoor walking	37 (55)	13 (54)	
Independent indoor walking	12 (18)	6 (25)	
Indoor walking with assistance	18 (27)	5 (21)	
Body composition			
Body mass index (kg/m^2^)	23.3 ± 3.8	25.6 ± 5.4	0.03
Metabolic rate (kcal)	1159 ± 135	1182 ± 138	0.48
Skeletal muscle mass (kg)	19.2 ± 3.8	19.8 ± 3.8	0.49
Body fat mass (kg)	17.5 ± 6.3	21.3 ± 8.5	0.02

Values are n (%) or means with standard deviations.

**Table 2 geriatrics-06-00003-t002:** Multivariate factors related to the rise in body temperature.

	Odds Ratio	95% CI	*p* Value
Age (years)	0.94	0.84–1.04	0.23
Male	0.99	0.38–2.63	0.99
Baseline BMI ≥ 25 kg/m^2^	0.49	0.18–1.35	0.17
Increase of metabolic rate	0.33	0.09–1.17	0.09
Increase of skeletal muscle mass	4.73	1.29–17.26	0.02
Increase of body fat mass	1.15	0.44–2.97	0.78

BMI: body mass index; CI: confidence interval.

## Data Availability

The data that support the findings of this study are available from the corresponding author upon reasonable request.

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
