# Peer review of "Effect of Exercise Training on Body Temperature in the Elderly: A Retrospective Cohort Study"

_geriatrics, 2021, doi:10.3390/geriatrics6010003_

Round 1

Reviewer 1 Report

In the present manuscript, the authors investigated the effect of exercise training on body temperature and the relationship between body temperature and body composition in the elderly. This study is interesting, the aim clear and results well described. However, different major implementations are necessary before publication.

Major Concern

Materials and methods

Study populations: there was some exclusion criteria? Moreover, was assessed baseline level of physical activity? Elderly were sedentary? Please specify them.

Exercise protocol

Exercise protocol description is scarce. The aim of this study is verify the effect of exercise training on body temperature, but exercise protocol is not explain enough.

Exercise intensity: what was the exercise intensity for both aerobic and anaerobic exercise?

Exercise protocol: please specify in more details the exercise protocol with intensity, set, repetitions, rest pause. Moreover, exercises were the same for all the subjects?

Exercise protocol progression: Throughout the 2-years, there was exercises modifications in terms of intensity and volume? There was difference between the two groups?

Results

Results about relationship between body temperature and change in body fat mass is missed. These results could be presented in the analysis. Moreover, body fat mass influence the change of body temperature?

The authors did not specify exercise intensity, volume, and their relative modifications throughout the 2-years. These information are essential, above all to explain more in depth the effect of exercise training on body composition.

Minor Concern:

Materials and methods

Study population

Lines 49-50: The authors declared that the elderly exercised for two year, but the data collection were performed between December 2015 and December 2018 (3-years). What is the main years of training? Are included subjects with more or less than 2 years of training? Please clarify this concept.

Evaluation of body temperature and body composition

Lines 55-57: How often body temperature was measured? Before each exercise sessions? Please, specify it.

Author Response

We greatly appreciate the Reviewer’s valuable comments/advice and careful reading of our paper. The comments have helped us improve the quality of the manuscript. The following is our point-by-point responses to the reviewer’s comments:

Major Concern

  1. Materials and methods

Study populations: there was some exclusion criteria? Moreover, was assessed baseline level of physical activity? Elderly were sedentary? Please specify them.

Our reply

Thank you for your important suggestion. We have added following sentences in the Methods section (P. 2) as suggested;

“Participants with interruption of exercise training during 2-year were excluded in this analysis.”

“Participants were categorized into 3 type of activity of daily living: 1) independent outdoor walking; 2) independent indoor walking, or 3) indoor walking with assistance.”

Additionally, the distribution of activity of daily living was added in the Table 1 and following sentence was added in the Results section (P. 3);

“The distribution of activity of daily living was comparable between the 2 groups.”

  1. Exercise protocol

Exercise protocol description is scarce. The aim of this study is verify the effect of exercise training on body temperature, but exercise protocol is not explain enough.

Exercise intensity: what was the exercise intensity for both aerobic and anaerobic exercise?

Exercise protocol: please specify in more details the exercise protocol with intensity, set, repetitions, rest pause. Moreover, exercises were the same for all the subjects?

Exercise protocol progression: Throughout the 2-years, there was exercises modifications in terms of intensity and volume? There was difference between the two groups?

Our reply

Thank you for your valued advice. We have added following sentences in the Methods section (P. 2-3) as suggested;

“Aerobic exercise intensity was determined according to heart rate and Borg’s scale 11-13 during exercise as referenced by previous reports [11]. Participants with indoor walking with assistance were excused from stair ascent and descent training.”

“Anaerobic exercise intensity was determined according to 40 to 60% of 1 repetition maximum as referenced by previous report [12]. All anaerobic exercise was attempted at 2 to 3 sets with 3 to 5 minutes interval among each set. Exercise intensity and volume were not changed throughout 2-year.”

  1. Results

Results about relationship between body temperature and change in body fat mass is missed. These results could be presented in the analysis. Moreover, body fat mass influence the change of body temperature?

The authors did not specify exercise intensity, volume, and their relative modifications throughout the 2-years. These information are essential, above all to explain more in depth the effect of exercise training on body composition.

Our reply

Thank you for your important comments. We have added change in body fat mass in the Figure 5 as suggested, resulting there was no significant difference in change in body fat mass between the 2 groups (47% vs. 25%, p = 0.06).

Moreover, increase of body fat mass was added into multivariate analysis as suggested (Table 2). As a result, increase of body fat mass was not significantly associated with the rise in body temperature (odds ratio 1.15, 95% confidence interval 0.44–2.97, p = 0.78).

Exercise intensity and volume were no changed throughout 2-year, therefore, we have compared with activity of daily living and body composition and body temperature including incidence of participants with increase of metabolic rate, skeletal muscle mass, body fat mass, and body temperature. As a results, there were no significant differences in incidence of participants with increase of body composition and body temperature among activity of daily living.

We have added following sentences in the Results section (P. 6);

3.4. Relationship between status of activity of daily living and body composition and body temperature

When participants were classified according to status of activity of daily living, incidence of increase in metabolic rate (56% vs. 44% vs. 39%, p = 0.36), skeletal muscle mass (40% vs. 44% vs. 35%, p = 0.82), body fat mass (66% vs. 50% vs. 48%, p = 0.25), and body temperature (80% vs. 78% vs. 96%, p = 0.19) were no significant differences among participants with independent outdoor walking, independent indoor walking, and indoor walking with assistance, respectively.”

Minor Concern:

  1. Materials and methods

Study population

Lines 49-50: The authors declared that the elderly exercised for two year, but the data collection were performed between December 2015 and December 2018 (3-years). What is the main years of training? Are included subjects with more or less than 2 years of training? Please clarify this concept.

Our reply

Thank you for your suggestion. We have changed following sentence in the Methods section to clarify this concept (P. 2);

“This single center retrospective cohort study included 91 consecutive elderly participants (≥ 70 years old) between December 2015 and December 2016. All participants who had exercise training twice a week for 2-year.”

  1. Evaluation of body temperature and body composition

Lines 55-57: How often body temperature was measured? Before each exercise sessions? Please, specify it.

Our reply

Body temperature was measured before each exercise sessions. Therefore, we have added following word in the Methods section (P. 2);

“Body temperature was measured before each exercise from participant’s forehead using a non-contact infrared thermometer (Thermo Phrase MT-500, NISSEI, Gunma, Japan) after 15 minutes rest in a room maintained at 20°C environmental temperature.”

In closing, let me thank you once again for your extremely cogent comments which have helped us improve the quality of our manuscript.

Reviewer 2 Report

General comments

In this paper, the authors summarize and describe the results obtained in a retrospective cohort study. They investigated the effects of a 2-year aerobic training program on body temperature ad tried to clarify the relationship between the increase in body temperature and body composition.

Body temperature rose significantly after exercise training in the lower body temperature group, and a positive correlation was observed between the amount of change in body temperature and baseline body temperature. The increase in skeletal muscle was an independent variable related to the rise in body temperature.

The authors concluded that exercise training could increase body temperature in the elderly, especially in the ones with lower baseline temperature. According to the authors, this effect may be beneficial for the immune response to external agents and cancer cells.

The paper is well written and easy to flow and understand. Although of limited novelty and based on a small number of volunteers, the topic has clinical and practical importance.

The paper's main glitch is the absence of a real control group, a problem inherent in eth retrospective nature of the experimental design. I also think that an equal increase in metabolic rate in the two groups of volunteers in contrast with the further increase of body temperature need further comments.

Specific comments (page, paragraph, line)

Page 2, para 3, line 56. Was the infrared thermometer ever validated?

Author Response

We greatly appreciate the Reviewer’s valuable comments/advice and careful reading of our paper. The comments have helped us improve the quality of the manuscript. The following is our point-by-point responses to the reviewer’s comments:

General comments

The paper is well written and easy to flow and understand. Although of limited novelty and based on a small number of volunteers, the topic has clinical and practical importance.

The paper's main glitch is the absence of a real control group, a problem inherent in eth retrospective nature of the experimental design. I also think that an equal increase in metabolic rate in the two groups of volunteers in contrast with the further increase of body temperature need further comments.

Our reply

Thank you for your important suggestion. We have added following sentences in the Discussion section (P. 6-7) as suggested;

“In this study, incidence of participants with increase in metabolic rate was comparable between participants with lower and normal body temperature group. In contrast, participants with increase in skeletal muscle mass was significantly higher in participants with lower body temperature compared to those with normal body temperature. Moreover, multivariate analysis indicated that increase in skeletal muscle mass was an independent variable related to rise in body temperature, whereas there was no significant association between increase in metabolic rate and rise in body temperature. These implies that skeletal muscle mass may be strongly attributable to rise in body temperature rather than metabolic rate. Regarding this point, further investigation is needed by clinical and experimental studies.”

“Three limitations of our study should be addressed. First, the study population was limited in number and retrospectively investigated. The findings of this study need to be confirmed prospectively in a larger population to determine the clinical significance of rise in body temperature in the elderly.”

“Third, this study did not include control group without underwent exercise training. Comparing change in body temperature and body composition between elderly participants with and without exercise training would be attributable to better understanding of association between exercise training and body temperature.”

Specific comments (page, paragraph, line)

Page 2, para 3, line 56. Was the infrared thermometer ever validated?

Our reply

Thank you for your advice. We have added following sentences in the Methods (P. 3) and Results (P. 4) section as suggested:

“To verify body temperature variation of a non-contact infrared thermometer, Bland-Altman plot was conducted. The upper and lower limits of agreement were set at 2 standard deviations from the mean.”

3.2. Body temperature variation

              To verify body temperature variation of a non-contact infrared thermometer, the difference between average body temperature and each body temperature in individual participant at baseline and at 2-year was evaluated (Figure 1). Bland-Altman plot revealed acceptable degree of agreement of body temperature variation using a non-contrast infrared thermometer was confirmed (bias 0.002, 95% confidence interval -0.006 to 0.010).”

Figure 1. Bland-Altman plot assessing body temperature variation of a non-contact infrared thermometer between average body temperature and each body temperature in individual participant at baseline and at 2-year. The upper and lower limits of agreement were set at 2 standard deviations from the mean (red lines). Black line indicates mean difference.”

Additionally, we have newly added Figure 1.

In closing, let me thank you once again for your extremely cogent comments which have helped us improve the quality of our manuscript.

Round 2

Reviewer 1 Report

The authors give exhaustive responsesto the questions.